# Prognosis Risk Model Based on Pyroptosis-Related lncRNAs for Gastric Cancer

**DOI:** 10.3390/biom13030469

**Published:** 2023-03-03

**Authors:** Min Jiang, Changyin Fang, Yongping Ma

**Affiliations:** Molecular Medicine and Cancer Research Center, Basic Medical College, Chongqing Medical University, Chongqing 400016, China

**Keywords:** GC, pyroptosis, LncRNAs, tumor microenvironment, prognosis, immunity, drug therapy

## Abstract

**Simple Summary:**

In this study, we aimed to determine the correlation between pyroptosis-related lncRNAs and gastric cancer prognoses. A novel predictive signature including six pyroptosis-related lncRNAs was established for the purposes of gastric cancer and immune status prognoses, which were achieved by using bioinformatics tools. After multiple validations, we confirmed that this signature possessed a good predictive performance. We found that high risk was associated with increased immune cell infiltration, increased immune function scores, and up-regulated expressions of immune checkpoints; in other words, the high-risk gastric cancer patients were more likely to benefit from the combination of immunotherapy and chemotherapy. Then, we performed quantitative reverse transcription polymerase chain reactions in order to verify the risk model. Further, the results indicated that pyroptosis-related genes play a crucial role in tumor progression and prognosis. In summary, the six pyroptosis-related lncRNAs in this study can be used as novel biomarkers for the prognosis and treatment of gastric cancer.

**Abstract:**

Gastric cancer (GC) is a malignant tumor with a low survival rate, high recurrence rate, and poor prognosis. With respect to this, pyroptosis is a type of programmed cell death that can affect the occurrence and development of tumors. Indeed, long non-coding RNAs (lncRNAs) were broadly applied for the purposes of early diagnosis, treatment, and prognostic analysis in regard to cancer. Based on the association of these three purposes, we developed a novel prognosis risk model based on pyroptosis-related lncRNAs (PRlncRNAs) for GC. The PRlncRNAs were obtained via univariate and multivariate Cox regression in order to build the predictive signatures. The Kaplan–Meier and gene set enrichment analysis (GSEA) methods were used to evaluate the overall survival (OS) and functional differences between the high- and low-risk groups. Moreover, the correlation of the signatures with immune cell infiltration was determined through single-sample gene set enrichment analysis (ssGSEA). Finally, we analyzed this correlation with the treatment responses in the GC patients; then, we performed quantitative reverse transcription polymerase chain reactions (qRT-PCRs) in order to verify the risk model. The high-risk group received a worse performance in terms of prognosis and OS when compared to the low-risk group. With respect to this, the area under the receiver operating characteristic curve (ROC) was found to be 0.808. Through conducting the GSEA, it was found that the high-risk groups possessed a significant enrichment in terms of tumor–immunity pathways. Furthermore, the ssGSEA revealed that the predictive features possessed strong associations with immune cell infiltration in regard to GC. In addition, we highlighted that anti-immune checkpoint therapy, combined with conventional chemotherapy drugs, may be more suitable for high-risk patients. The expression levels of LINC01315, AP003392.1, AP000695.2, and HAGLR were significantly different between the GC cell lines and the normal cell lines. As such, the six PRlncRNAs could be regarded as important prognostic biomarkers for the purposes of subsequent diagnoses, treatments, prognostic predictions, and the mechanism research of GC.

## 1. Introduction

Despite the steady decline in the global incidence of GC over the past few decades, GC remains the most significant cancer worldwide. Indeed, as of 2020, GC was, statistically, ranked fifth in terms of incidence and fourth in terms of mortality worldwide. This is with more than 1 million new patients and about 769 thousand deaths in one year [1]. Furthermore, due to the high heterogeneity and the deficiency with respect to the specific symptoms of GC, it is often undiscovered until the condition of the affected patient deteriorates [2,3]. At present, the first choice of GC patients with respect to treatment is still surgical resection combined with adjuvant chemoradiotherapy [4]. However, the high recurrence rate and low survival rate of patients with advanced gastric cancer still led to a large number of deaths [5]. Although the combined application of chemo-immunotherapy has significantly prolonged the OS of GC patients, the development of multidrug resistance (MDR) of GC cells often leads to poor patient prognoses.

Pyroptosis is a multifactorial and catalyzed form of cell death. In 2018, the Cell Death Nomenclature Committee (NCCD) proposed to define cell pyroptosis as a type of regulatory cell death (RCD), which is usually caused by the activation of inflammatory caspase. The activation of the gasdermin (GSDM) protein family members, as well as the formation of plasma membrane pores on the cell membrane surface, is a key step in the process [6]. Hence, pyroptosis is also defined as GSDM-mediated programmed necrosis [7]. The cleavage of GSDMs leads to continuous cell expansion until rupture, and the release of cell content causes a robust inflammatory response [8]. The major inducers are found in the canonical caspase-1 inflammasome or caspase-4, -5, and -11, which are activated by cytosolic lipopolysaccharides [9]. Recent studies report that pyroptosis can impact all of the stages of carcinogenesis, and inducing pyroptotic cellular death could be a promising therapeutic option for managing and regulating multiple cancers [10,11]. It was reported that the down-regulation of gasdermin D (GSDMD) protein expression, a key factor triggering pyroptosis, could induce the proliferation of GC cells by regulating cell cycle-related proteins [12]. After chemotherapy, the drug-induced cleavage of GSDME by caspase-3 would mediate apoptosis into pyroptosis in some cancer cells expressing GSDME [13]. In addition, studies have found that a variety of drugs play anti-tumor effects through pyroptosis-related pathways [14,15].

LncRNAs are molecules with a transcript length of more than 200 nts. They do not encode proteins but only participate in their regulation in the form of RNA. Many studies have revealed that lncRNAs are an essential part of the normal functioning of biological mechanisms and are involved in a variety of pathophysiological activities, including psychiatric disorders, tumors, and cardiovascular diseases, as well as aging [16,17,18,19]. More importantly, lncRNAs possess an abnormal expression in GC, which can affect the progression, metastasis, treatment, and prognosis of gastric cancer through multiple pathways [20,21,22,23]. Therefore, lncRNAs act as biomarkers for the purposes of early GC detection [24,25]. Moreover, recent studies have found that certain lncRNAs may regulate pyroptosis in order to affect tumor proliferation, invasion, and metastasis [26,27]. Meanwhile, increasing evidence shows that LncRNAs and related mRNAs can induce drug resistance in gastric cancer [28,29,30], indicating the potential clinical significance of ncRNAs as a new therapeutic target and prognostic biomarker for gastric cancer.

Having said this, the prognostic significance, as well as the diagnostic and therapeutic value of PRlncRNAs in GC, have rarely been studied. It is still unclear whether they could be a prognostic indicator of GC or whether they are related to the immune functions of GC and drug treatments. Therefore, we established a prediction signature based on PRlncRNAs in order to provide new treatment guidance for GC, ameliorate the worst outcome of GC, and further explore its working mechanism. We also employed two external cohorts to validate the prognostic, predictive ability of the risk model. Then, we combined predictable clinicopathological features and risk scores to construct an efficient nomogram to predict the survival rate of GC patients. Furthermore, the molecular characteristics, clinical significances, tumor microenvironment, immune checkpoint profiles, and benefits of chemotherapy regarding the PRlncRNA signature were also explored.

## 2. Materials and Methods

### 2.1. Patients and Datasets 

The RNA-seq (i.e., 32 normal samples and 375 tumor samples) and the clinical data of GC were acquired from the TCGA website (https://portal.gdc.cancer.gov/, accessed on 13 January 2022), based on the gene annotated GTF files from the GENCODE website (https://www.gencodegenes.org/human/, accessed on 17 June 2021). We identified 14,142 lncRNA and 19,658 protein-coding genes. To fully verify the differential expression of the LncRNAs in gastric cancer and normal tissues, the genotypic tissue expression data of 359 normal gastric tissues were downloaded from GTEx website (https://gtexportal.org/home/, accessed on 21 February 2023). By taking the intersection of identified protein-coding genes and 255 pyroptosis-related genes, which were downloaded from GeneCards (https://www.genecards.org/, accessed on 25 January 2022), 239 pyroptosis-related genes of GC patients were determined. Moreover, Cytoscape (version 3.6.1, https://cytoscape.org/, accessed on 9 March 2022) was used in order to visualize the co-expression networks between PRlncRNAs and genes. 

### 2.2. Establishment and Validation of the PRlncRNAs Risk Model 

Pearson correlation analysis was performed using R software (version 4.1.3, https://www.r-project.org/, accessed on 30 January 2022) in order to calculate the correlations between the 14,142 lncRNAs and 239 pyroptosis-related genes. The 1310 lncRNAs associated with pyroptosis were identified with correlation coefficients |R2| > 0.3 and *p* < 0.001. Patients with an overall survival of <30 days were excluded. In addition, we successively used univariate and multivariate Cox regression analysis in order to obtain six selected PRlncRNAs, which were then used to confirm the prediction feature. The formula was as follows: Risk score = $\sum_{i=1}^{n}\left(Coefi\;\times \;Xi\right)$. Coef: coefficient value; X: expression of lncRNAs. This PRlncRNAs risk model was further validated in two GEO cohorts (GSE15459 and GSE62254). Subsequently, the decision curve analysis (DCA) was performed to evaluate the intended clinical effectiveness of this model with two published related models [31,32].

### 2.3. Prediction of OS in GC Patients by Establishing A Prognostic Nomogram 

A nomogram was built using the rms package in R 4.1.3 to predict the one-, three-, and five-year survival rates of GC patients by combining risk scores with the clinicopathologic features of: age, sex, grade, and disease stage. Meanwhile, a calibration curve tested the agreement between prediction and reality.

### 2.4. Differences in Functional Enrichment and Immune Infiltration within the Two Risk Groups

We grouped the GC patients into high- and low-risk groups according to their median risk score. We analyzed the significantly enriched pathway genes through GSEA4.1.0 (https://www.gsea-msigdb.org/gsea/index.jsp, accessed on 5 April 2022). The threshold criteria were set with a statistical significance of *p* < 0.05 and FDR < 0.25. Subsequently, we used the "GSVA" package in order to perform the ssGSEA [33].

### 2.5. Evaluation of Immunotherapy and Drug Therapy

Conventional chemotherapy drugs for GC were used to calculate the half-maximal inhibitory concentration (IC50). Then, this was compared between the two groups through a Wilcoxon signed-rank test in R 4.1.3 package pRRophetic.

### 2.6. Cell Culture and qRT-PCR

Normal gastric epithelial cell lines (GES-1) and 5 human gastric cancer cell lines (AGS, MKN45, MKN28, HGC27, and SGC-7901) were preserved in our laboratory. These cells were cultured in an RPMI-1640 medium (Gibco, Billings, MT, USA) containing a 10% fetal bovine serum (FBS, Gibco, Billings, MT, USA) of penicillin and streptomycin (Gibco, Billings, MT, USA, 1 U/mL and 0.1 mg/mL, respectively), which were incubated at 37 °C in a humidified atmosphere of 5% CO_2_. The total RNA was isolated from the cells by using an SV Total RNA Isolation System (Promega, Madison, WI, USA). The quality and concentration of RNA samples were examined by using NanoDrop™ 2000/2000c Spectrophotometers (Thermo Fisher Scientific, Waltham, Massachusetts, USA). Moreover, the RNA (2 μg) was reversely transcribed into cDNA using a GoScript Reverse Transcription System (Promega, Madison, WI, USA), according to the manufacturer’s instructions. The quantitative RT–PCR was performed using a TaKaRa SYBR® Premix Ex Taq™ (TaKaRa, Dalian, China) on a CFX Connect Real-Time PCR Detection System (Bio-Rad, Hercules, CA, USA). The volume was adjusted by: 20 μL of the total reaction volume; 10 μL of the SYBR Premix Ex Taq II (2×); 2 μL of cDNA; 0.8 μL of the upstream and downstream primers; and the addition of sterile purified water. The amplification conditions were: 95 °C for 30 s, 95 °C for 5 s, 60 °C for 30 s, and 40 cycles. Furthermore, 2^–ΔΔCq^ was used to analyze the data, and the experiment was repeated three times. All primers used in the PCR are listed in Table 1. Lastly, the expression levels of the risk lncRNAs were normalized to the expression levels of GADPH. 

### 2.7. Statistical Analysis 

The latest version of R software (version 4.1.3) was used as the statistical analysis tool. The final PRlncRNA, as well as the relationship between PRlncRNA and OS, were screened using univariate and multivariate Cox regression analysis. The OS of the patients in the two groups were analyzed by the Kaplan–Meier method and log-rank test. In addition, the ROC curve was drawn by the survival ROC package. Lastly, the ssGSEA was performed via the GSVA package. 

## 3. Results

### 3.1. Identification of Significant Prognostic Values of PRlncRNAs in GC 

A complete workflow for our research process is depicted in Figure 1. By analyzing the correlation coefficient of the pyroptosis-related coding genes (mRNAs) and lncRNAs, we identified 1310 PRlncRNAs. Through the univariate Cox regression analysis, we obtained 28 PRlncRNAs for the GC patients with a hazard ratio (HR) < 1, which means low risk, and an HR > 1, which means high risk (Figure 2A). After completing the multivariate Cox regression analysis, six PRlncRNAs were confirmed in order to establish a forecasting model; namely, LINC01315, AL161785.1, AP003392.1, AP000695.2, HAGLR, and AL590666.2 (Table 2). The expressions of the six PRlncRNAs in GC are shown in Figure 2B. Then, we compared the expression level of the LncRNAs in gastric cancer and normal adjacent tissues. We found that there were significant differences in the expression levels of 6 LncRNAs regardless of the single TCGA data set (Tumer = 375, Normal = 32) or TCGA combined with GTEx data set (Tumer = 375, Normal = 391). Among them, LINC01315, AP003392.1, AL590666.2, and HAGLR were highly expressed in gastric cancer tissues, while the expression levels of AP000695.2 and AL161785.1 in gastric cancer tissues were lower than those in normal tissues (Appendix A). The lncRNAs were visualized by Cytoscape and the ggalluvial in the R software package. The 16 pairs of the lncRNA-mRNA co-expression network are displayed in Figure 2C (|R2| > 0.3 and *p* < 0.001). Moreover, AP003392.1 possessed a co-expressive relationship with nine pyroptosis-related genes (i.e., NLRP1, CARD8, DPP8, MALAT1, AGER, KLF3-AS1, MEG3, CHMP4A, and TIRAP). Furthermore, AP000695.2 possessed a co-expressive relationship with two pyroptosis-related genes (IL18 and ANXA2). Indeed, AL590666.2 possessed a co-expressive relationship with two pyroptosis-related genes (E2F4 and CDC37). Similarly, LINC01315 was co-expressed with ASIC1, AL161785.1 with FNDC4, and HAGLR was co-expressed with GSDMA. However, LINC01315, AP003392.1, and AL590666.2 were found to be protective factors. Further, AL161785.1, AP000695.2, and HAGLR were established as the risk factors (Figure 2D). The risk score = (−0.448 × LINC01315 exp) + (0.581 × AL161785.1 exp) + (−0.673 × AP003392. 1 exp) + (0.564 × AP000695.2 exp) + (−0.449 × HAGLR exp) + (−0.217 × AL590666.2 exp), exp: expression.

### 3.2. Exploring the Value of Six Prlncrnas as Independent Risk Models in Terms of Predicting the Outcome of GC Patients

Patients were divided into high- and low-risk groups in accordance with the median calculated risk score. The results indicated, as determined by the Kaplan–Meier method’s analysis, that the OS of the high-risk group was significantly shorter than that of the low-risk group (Figure 3a, *p* < 0.001). The risk scores of the two groups are shown in Figure 3b. Moreover, the mortality of the patients was positively correlated with the risk score (Figure 3c). The expressions of AL161785.1, AP000695.2, and HAGLR were up-regulated in the high-risk group, whereas in regard to the low-risk group, the LINC01315, AP003392.1, and AL590666.2 were up-regulated (Figure 3d). The value of a predictive characteristic in terms of a GC prognosis was assessed by conducting a Cox regression analysis. Furthermore, by conducting a univariate Cox regression analysis, it was shown that the risk score, age, and tumor size, nodal status, and metastases(TNM) stage, as well as the T and N stage of GC patients, were all closely correlated with the OS in GC patients (Figure 4A). As per the multivariate Cox regression, only age and risk score were closely correlated with the OS (Figure 4B). Indeed, the area under the ROC curve (AUC) of the risk score was 0.808, thereby indicating that the predictive signature possessed a good predictive performance (Figure 4C). Furthermore, the AUCs for the one-, three-, and five-year survival were 0.734, 0.709, and 0.775, respectively (Figure 4D). DCA is a method for evaluating and comparing multiple clinical prediction models. The decision curves showed the highest net benefit for the PRlncRNAs’ signature (“This model”) compared with the default strategies (“All” and “None”) and clinical traits with prognostic significance (“854785” represented the model from [31]; “816153” represented the model from [32]) (Figure 4E–G). The results showed that the N and T stage, TNM stage, age, and vital status were significantly different between the two groups (Figure 5).

In order to further assess the effect of the predictive features, we created a nomogram integrating the risk score and clinical factors of age, stage, and grade for comprehensively predicting the GC patients’ OS at 1, 3, and 5 years (Figure 6A). We showed the use of the nomogram in the supplementary document, and compared the efficiency of the nomogram made by the model with the traditional prognostic indicators through DCA (Appendix A). Our nomogram displayed better accuracy in predicting both short- and long-term survival. In addition, the calibration plot of the nomogram for 1, 3, and 5 years showed optimal agreement between the prediction by the nomogram and the actual observation outcome (Figure 6B–D).

### 3.3. Assessing the Effect of Different Clinicopathological Variables on the Performance of Predictive Signatures in GC Patients

In order to determine the influence of the different clinicopathological variables with respect to the predictive signature efficiency in the GC patients, the GC patients were thus grouped according to their clinical manifestation. In all the groups, the OS of the low-risk group was significantly longer than that of the high-risk group (Figure 7). Furthermore, the results suggested that the predictive signature remained valid, as well as independent of the clinicopathological variables. 

### 3.4. Verification of the Risk Model

A total of 294 GC samples were randomly assigned to the training and testing sets (*n* = 147) in order to verify the applicability of the predicted signatures based on the entire TCGA dataset. The clinical statistics of the 294 patients are shown in Table 3. The results of the two sets were aligned with the entire set, thus showing that the low-risk group possessed a higher OS rate than that of the high-risk group (Figure 8A, *p* = 5.03 × 10^−8^; Figure 8C, *p* = 1.805 × 10^−4^). The AUCs of the 5-year survival group with respect to the training and testing sets were 0.795 and 0.74, respectively—thereby demonstrating good prediction performance (Figure 8B,D). To evaluate the accuracy of the model’s prediction results, the predictive ability of the risk model was also validated in GSE62254 and GSE15459. The AUCs for predicting OS at 1, 3, and 5 years were 0.622, 0.705, and 0.714 in GSE15459 and 0.682, 0.617, and 0.630 in GSE62254, respectively (Figure 8E–H). In line with the TCGA cohorts, our risk model could independently predict the prognosis of GC patients in two external test cohorts.

### 3.5. Differences in Immune Cell Infiltration and Immune-Related Pathways between the Two Groups

The differences regarding the tumor immune cell infiltration and immune function were analyzed with the use of ssGSEA between the different risk groups. In terms of the immune cells, there were significant differences between the two risk groups in the activated dendritic cells (aDCs), B cells, CD8+ T cells, DCs, immature dendritic cells (iDCs), macrophages, mast cells, neutrophils, plasmacytoid dendritic cells (pDCs), T helper cells, tumor-infiltrating lymphocytes (TILs), and regulatory T cells (Tregs) (Figure 9A). Regarding the immune function score, the scores of the antigen-presenting cell (APC) co-inhibition, APC co-stimulation, chemokine receptor (CCR), checkpoint, human leukocyte antigen (HLA), para-inflammation, type II interferon response, T cell co-stimulation, and T cell co-inhibition in the high-risk group were significantly higher than those found in the low-risk group (Figure 9B). In other words, the immune function of the high-risk groups was found to be more active. Given the importance of current immunotherapy for patients with GC, we further analyzed the distribution of immune checkpoints in the two risk groups. The results indicated that most of the immune checkpoints were significantly up-regulated in the high-risk group, whereas only TNFRSF25 and TNFRSF14 were significantly up-regulated in the low-risk group (Figure 9C).

### 3.6. Functional Enrichment Analysis Conducted via GSEA between Different Risk Patients 

The possible discrepancies in the signaling pathways between the two risk groups were investigated via the use of GSEA. The results showed that the extracellular matrix (ECM) receptor interaction, cytokine receptor interaction, lysosome, complement and coagulation cascades, calcium signaling pathways, and cell adhesion molecule cams were significantly enriched in the high-risk group. In contrast, the RNA degradation, spliceosome, base excision repair, basal transcription factors, and aminoacyl tRNA biosynthesis were significantly enriched in the low-risk group (Figure 10A). 

### 3.7. Relationships between the Predictive Signature and Treatment of GC 

Several commonly used GC chemotherapy drugs were selected in order to evaluate the correlation between chemotherapy and the predictive signature, as well as to then develop better treatment plans. The results suggested that Bryostatin.1, Cisplatin, Dasatinib, Docetaxel, Gemcitabine, Parthenolide, Pazopanib, Rapamycin, Sunitinib, Temsirolimus, and Vinblastine possessed lower IC50s in the high-risk group, while Mitomycin C possessed higher IC50s in the low-risk group (Figure 10B–M).

### 3.8. Differential Expression of the Six PRlncRNAs 

Moreover, we selected four PRlncRNAs (LINC01315, AP003392.1, AP000695.2, and HAGLR) with known sequences in the model to validate the expression of PRlncRNAs in the gastric epithelial cell line, GES-1, and the GC cell lines, AGS, MKN45, MKN28, HGC27, and SGC-7901, by qRT-PCR. As shown in Figure 11A–D, we could observe that LINC01315, AP003392.1, and HAGLR were significantly upregulated in the GC cell lines compared to those in the GES-1 cells. Meanwhile, the expressions of AP000695.2 were significantly lower in the GC cell lines, and it is consistent with the results of our bioinformatics analysis and lays a foundation for our further institutional research.

## 4. Discussion

GC often occurs in atrophic gastritis and intestinal metaplasia; furthermore, it mostly originates from the chronic infection of the gastric mucosa [34]. As a multifactorial disease, GC possesses the characteristics of a low OS rate, high recurrence rate, and poor prognosis [35]. Due to the issue of poor prognosis, targeted therapy and prognostic evaluation are particularly important in order to improve the prognosis and life happiness index of GC patients. Therefore, it is necessary to develop a high-quality and powerful prognostic risk assessment model for GC.

Pyroptosis usually involves the occurrence and development of malignant tumors; further, this issue may be a double-edged sword with respect to the pathogenesis of tumors. Pyroptosis activates multiple signaling pathways and releases inflammatory mediators in order to induce tumorigenesis, as well as to cause chemotherapy resistance [36]. In addition, pyroptosis can inhibit tumor progression through cell death [37]. 

As a research hotspot and difficulty in the era of biological genomics, many studies have shown that tumor occurrence, metastasis, and tumor stage are closely related to the abnormal expression and mutation of lncRNAs [38,39]. Interestingly, recent studies have reported that lncRNAs also have some influence on tumor recurrence [40,41]. Therefore, due to their specific phrase in certain types of cancer, lncRNAs could represent a potential new class of prognostic, diagnostic, and therapeutic targets for treating cancer.

The roles of lncRNAs and pyroptosis in the progression of GC have been described above in this study. With the development of biomedicine, PRlncRNAs are widely used in the diagnosis and treatment of tumors, including GC [31,32,42,43]. Therefore, it is necessary to generate a PRlncRNA predictive signature in order to predict the prognosis of GC. In this study, we identified six PRlncRNAs (LINC01315, AL161785.1, AP003392.1, AP000695.2, HAGLR, and AL590666.2) and included them in the predictive risk model. Previous studies have shown that AP003392.1 was used to establish a ferroptosis-related risk model in order to predict the prognosis of GC [44]. Similarly, Wang et al. reported that AP000695.2 acts as a novel prognostic biomarker and regulates the cell growth and migration of lung adenocarcinoma [45]. As such, HAGLR and LINC01315 have become diagnostic and prognostic markers for various cancers due to their close relationship with malignant tumors. In addition, HAGLR sponges, miR-338-3p, could promote 5-Fu resistance in gastric cancer by targeting the LDHA-glycolysis pathway [46]. Instead, the overexpression of LINC01315 predicted a worse outcome of triple-negative breast cancer [47]. Additionally, LncRNA AL161785.1 was used to construct a prognostic model of gastric adenocarcinoma related to glycolysis [48]. The previous findings indirectly confirmed the reliability of the prediction model in this study. Having said that, AL590666.2 was first identified as a new biomarker in order to predict the prognosis of GC patients. We also found that the mRNAs (i.e., the encoded NLRP1, CARD8, DPP8, MALAT1, AGER, KLF3-AS1, MEG3, CHMP4A, IL18, E2F4, FNDC4, GSDMA, CDC37, ANXA2, TIRAP, and ASIC1) were significantly co-expressed with these lncRNAs. Moreover, NLRP1 and CARD8 are inflammasome-related pattern recognition receptors; further, they activate inflammatory cytokines and induce pyroptosis in order to respond to intracellular risk-related signals [49]. The inhibition of Dpp8/9 activates the Nlrp1b and CARD8 in inflammasomes and results in pro-caspase-1-mediated pyroptosis [50]. Moreover, ANXA2 can promote the progression and metastasis of GC through the EphA2-YES1-ANXA2 signaling pathway [51]. ASIC1 promotes NLRP3 inflammasome activation and IL-1β release in order to induce pyroptosis [52]. FNDC4 inhibits pyroptosis by blunting the inflammasome activation of GSDMD-processing and the IL-1β release [53]. In addition, GSDMA is an essential mediator in the process of cell pyroptosis [54]. Here, we built a predictive model based on six pyroptosis-related lncRNA signatures for the purposes of GC prognosis. The prognostic pyroptosis signature can successfully categorize patients into subtypes with different survival outcomes. The predictive signature is established in TCGA dataset and verified by GEO external dataset. At the same time, it is screened by large-scale co-expression of LncRNA and the genes related to pyroptosis. The strict screening criterion is correlation coefficients | R2 | > 0.3 and *p* < 0.001. Big data analysis of bioinformatics and in vitro cells were carried out to verify the differential expression of the model between normal and cancer tissues, and the prospect of the model in tumor immune microenvironment and immunotherapy was deeply analyzed. Due to the differences in screening methods, identification criteria, and sources of data sets, the genes we identified are different from those of existing models, but the verification results show some advantages of the model The result of internal and external verification showed that the predictive signature possessed an excellent performance and was not disturbed by clinicopathological variables. Moreover, we also compared the effectiveness of the existing prognosis model with ours through the DCA decision curve, and the results confirmed that the model constructed in this study is obviously superior to other models in predicting survival. Then, we constructed the nomograph based on the multi-factor regression analysis. According to the contribution of each influencing factor in the model to the outcome variable, the predicted value of the individual outcome event can be calculated intuitively so as to facilitate the evaluation of patients.

Regarding the results of ssGSEA, we found that several pathways were positively associated with risk scores, including ECM receptor interaction, lysosome, cytokine receptor interaction, cell adhesion molecules, and calcium signaling pathways. Previous studies have showed that αvβ6, as an abnormal expression of the ECM receptor in GC, is associated with decreased survival rate and is recommended as a prognostic marker for early tumors [55]. Other evidence suggests that cell adhesion molecules on malignant cells are involved in invasion, metastasis, and immunosuppression [56,57]. The above results indicate that high-risk patients are strongly associated with tumor-immune pathways, and more of them are immunosuppressive pathways. 

Through conducting the ssGSEA, it was revealed that the infiltration of Treg cells, mast cells, and macrophages had increased in the high-risk group. With respect to this, recent studies have indicated that mast cell infiltration in GC patients is increased and positively related to tumor progression, thus indicating a poor prognosis [58]. The high infiltration of Treg cells and Treg-mediated immune escape are correlated with poor survival in regard to various cancers [59]. In this study, the high-risk group showed a stronger immune response to HLA, para-inflammation, Type I Interferon (IFN), and Type II IFN; in other words, the decline in anti-tumor immune function may lead to a poor prognosis. 

In order to further explore the guiding role of the model in the chemotherapy and immunotherapy of patients with different subtypes of gastric cancer, we analyzed the difference in the expression of the immune checkpoints and the sensitivity to general chemotherapy drugs in different groups. According to our results, most immune checkpoints are highly expressed in high-risk groups such as CD27, CD28, and CD40, indicating that immune checkpoint inhibitors may be more beneficial to high-risk groups for the up-regulation of immune checkpoint-related genes [60,61,62]. Meanwhile, we highlighted the fact that high-risk patients exhibited higher sensitivities to conventional chemotherapy drugs, such as Bryostatin.1, Cisplatin, Dasatinib, Docetaxel, Gemcitabine, Parthenolide, Pazopanib, Rapamycin, Sunitinib, Temsirolimus, and Vinblastine, the only exception being Mitomycin C. This conclusion is also largely consistent with clinical dosing habits, where surgery combined with radiotherapy is usually used for patients evaluated for poor prognosis [63]. Therefore, the present model for patient stratification is more applicable to clinical treatment. Taken together, high-risk patients, after assessment by this model, may be better candidates for the combination of immunotherapy and chemotherapy. In addition, we detected the expression levels of these four PRLs in the gastric epithelial cells and gastric cancer cell lines via qRT-PCR, which increased the credibility of our study. The results are consistent with our analysis, and in future studies, we intend to use AP003392.1 and AP000695.2 as targets to investigate their mechanistic pathways in gastric cancer development in depth, analyze their potential use as diagnostic markers, and explore the value of prediction models for recurrence risk assessment. 

In summary, a newly pyroptosis-related long non-coding RNA prognostic model has been constructed, which may provide a better treatment strategy and clinical management for GC. The six PRlncRNAs’ signatures showed high stability through multiple validations on different data platforms. Compared with other models in the same period, it has certain advantages in prediction validity. Through exploration of the multi-omics alterations, immunological profiles, and pharmacological landscape of the six PRlncRNAs, the accuracy and clinical applicability of the model were proved. Nevertheless, our research possesses certain limitations. The actual application effect of the prediction model in clinical practice is still unknown, and its internal mechanism remains to be further studied, which are two topics we need to explore in the future.

## 5. Conclusions

In this study, we identified six PRlncRNAs that were associated with GC prognosis and then built a prognosis-predictive risk model. The model possesses a good predictive performance and may become a neoadjuvant tool for the purposes of the clinical prognostic analysis and treatment of GC. These six PRlncRNAs are potential prognostic biomarkers and may provide a reference for the study of GC pyroptosis.

## Figures and Tables

**Figure 1 biomolecules-13-00469-f001:**
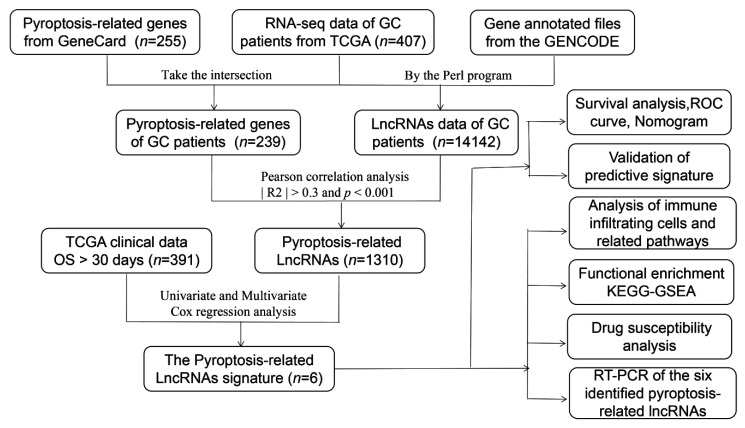
Workflow of the prognostic risk model analysis.

**Figure 2 biomolecules-13-00469-f002:**
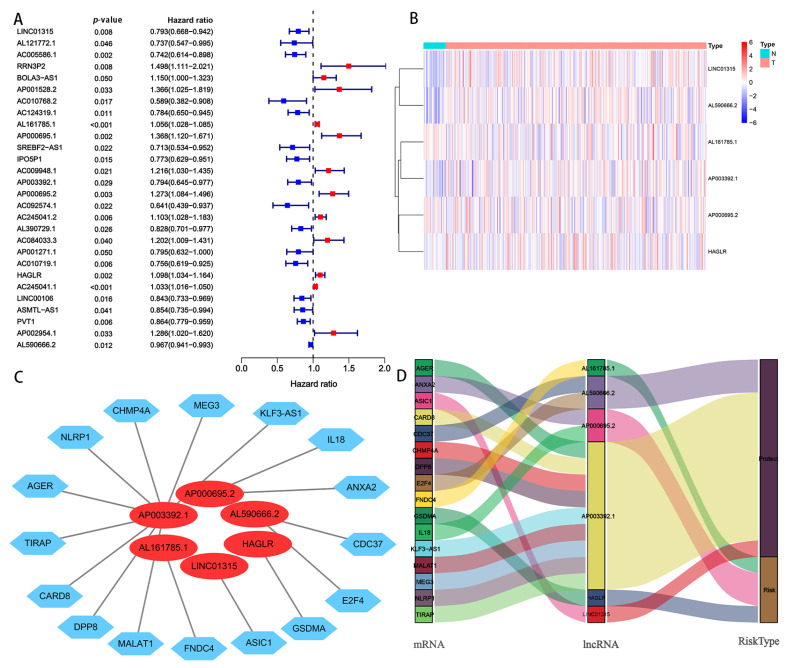
Screening of PRlncRNAs, which were significantly associated with a prognosis of GC. (**A**) A univariate Cox proportional hazards analysis revealed the HR (95% CI) and *p*-value of the selected lncRNAs in the forest plot. (**B**) Heat map of the differential expression of six PRlncRNAs with a prognostic value in regard to gastric cancer and normal tissues. (**C**) The prognostic pyroptosis-related lncRNA co-expression network. (**D**) The visualized Sankey diagram. N: nomal; T: tumer; NLRP1, CHMP4A, MEG3, KLF3-AS1, IL 18, ANXA2, CDC37, E2F4, GSDMA, ASIC1, FNDC4, MALAT1, DPP8, CARD8, TIRAP, AGER: Pyroptosi*s*-related mRNAs.

**Figure 3 biomolecules-13-00469-f003:**
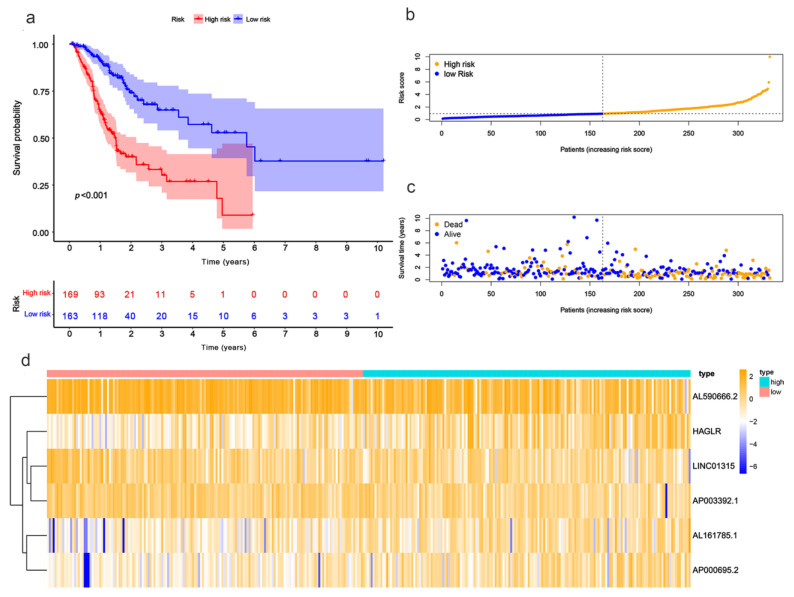
Evaluation of the prognostic value regarding the six PRlncRNAs’ risk models. (**a**) Kaplan–Meier OS curves with respect to the high- and low-risk groups. (**b**) The risk curves, as based on the patient risk scores. (**c**) Scatter plot, which was established according to the patients’ survival statuses. The blue dots represent survival, and the orange dots represent death. (**d**) A heatmap that shows the differential expression levels of PRlncRNAs with respect to the high- and low-risk groups.

**Figure 4 biomolecules-13-00469-f004:**
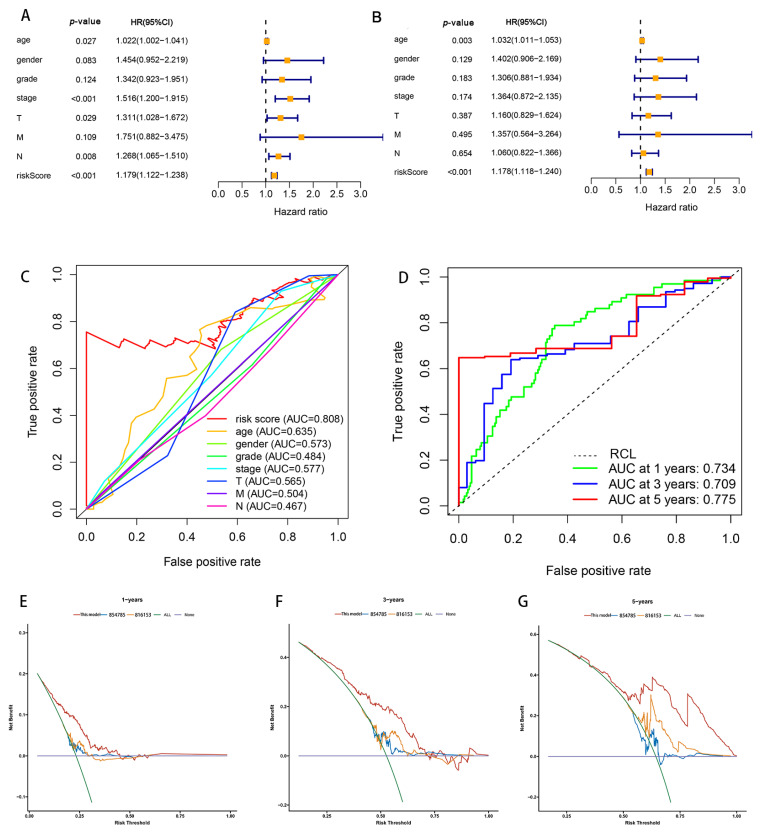
Correlation of risk models regarding the six PRlncRNAs with a prognosis in GC. (**A**) The univariate Cox regression analysis. (**B**) The multivariate Cox regression analysis. (**C**) The ROC curves, which showed the AUC for the risk scores and clinicopathological variables. The clinicopathological variables were as follows: age, sex, grade, TNM stage, T, and tumor. Abbreviations—M: metastasis and N: lymph node. (**D**) The ROC curves and AUCs for the one-, three-, and five-year survival rates, as per the risk model. (**E**–**G**) Decision curve analysis (DCA) curves for overall survival at 1, 3, and 5 years to evaluate the efficacy of different clinical models, and the red line represents this model; the orange line represents the model from [31]; the blue line represents the model from [32]. RCL: random guessing line, the baseline of a ROC curve.

**Figure 5 biomolecules-13-00469-f005:**
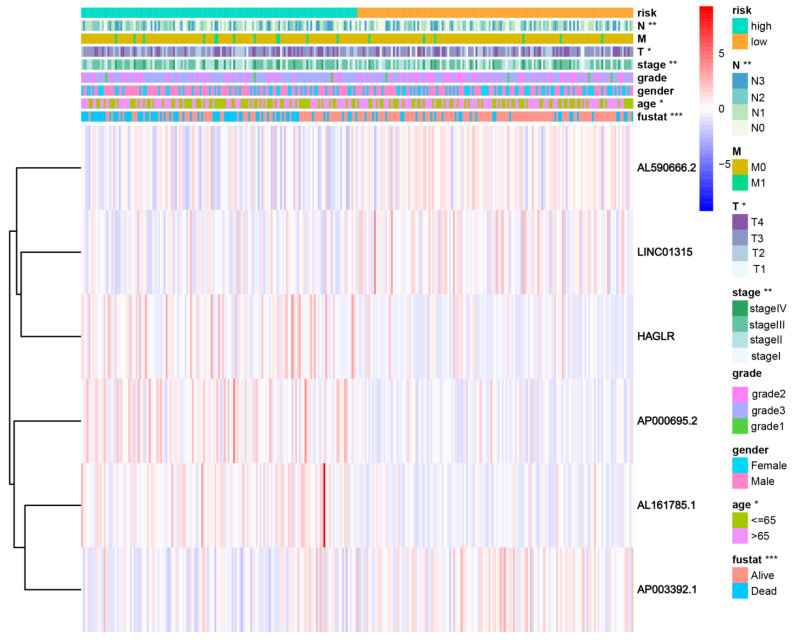
The connection with the prognostic PRlncRNAs, as well as the clinicopathological factors with the risk groups. * *p* < 0.05; ** *p* < 0.01; and *** *p* < 0.001. Fustat: vital status.

**Figure 6 biomolecules-13-00469-f006:**
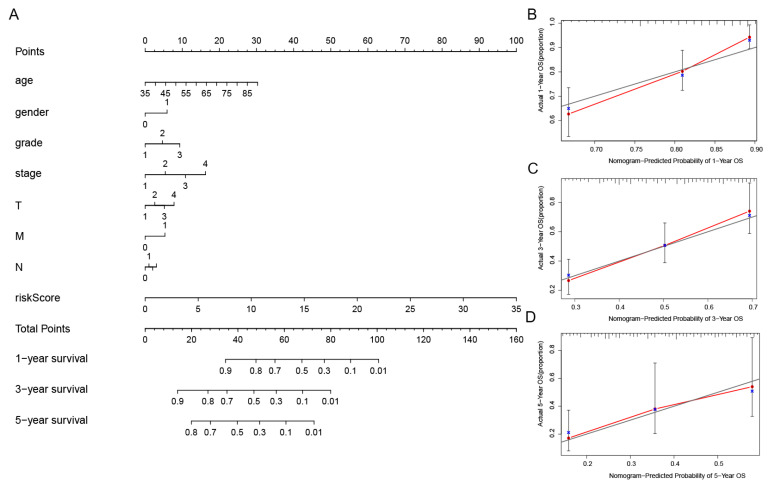
Establishment of the nomogram of the predicted signatures, as well as the verification of its reliability. (**A**) The one-, three-, and five-year OS in patients with GC, as forecasted by the nomogram containing the clinicopathological variables and risk scores. (**B**–**D**) The calibration curves that were used in order to explore whether the actual OS rates were consistent with the predicted survival rates. Perfect prediction would correspond to the grey line. The blue dots represent the actual incidence. The red line is the predicted fitting line obtained by bias correction.

**Figure 7 biomolecules-13-00469-f007:**
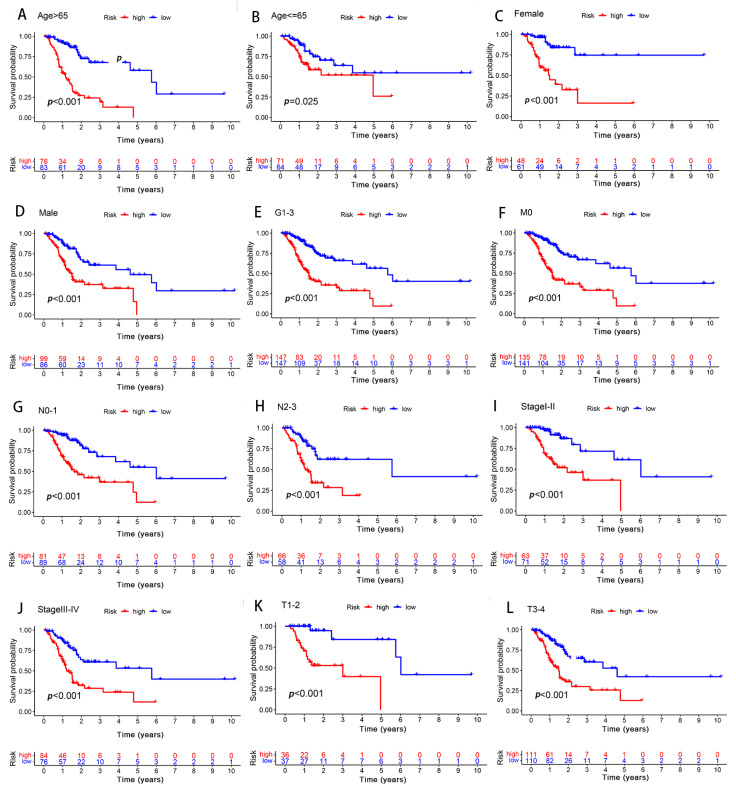
Differences in survival between the high- and low-risk groups, as grouped based on the different clinicopathological variables. (**A**,**B**) Age. (**C**,**D**) Sex. (**E**) Grade. (**F**) M stage. (**G**,**H**) N stage. (**I**,**J**) stage. (**K**,**L**) T stage.

**Figure 8 biomolecules-13-00469-f008:**
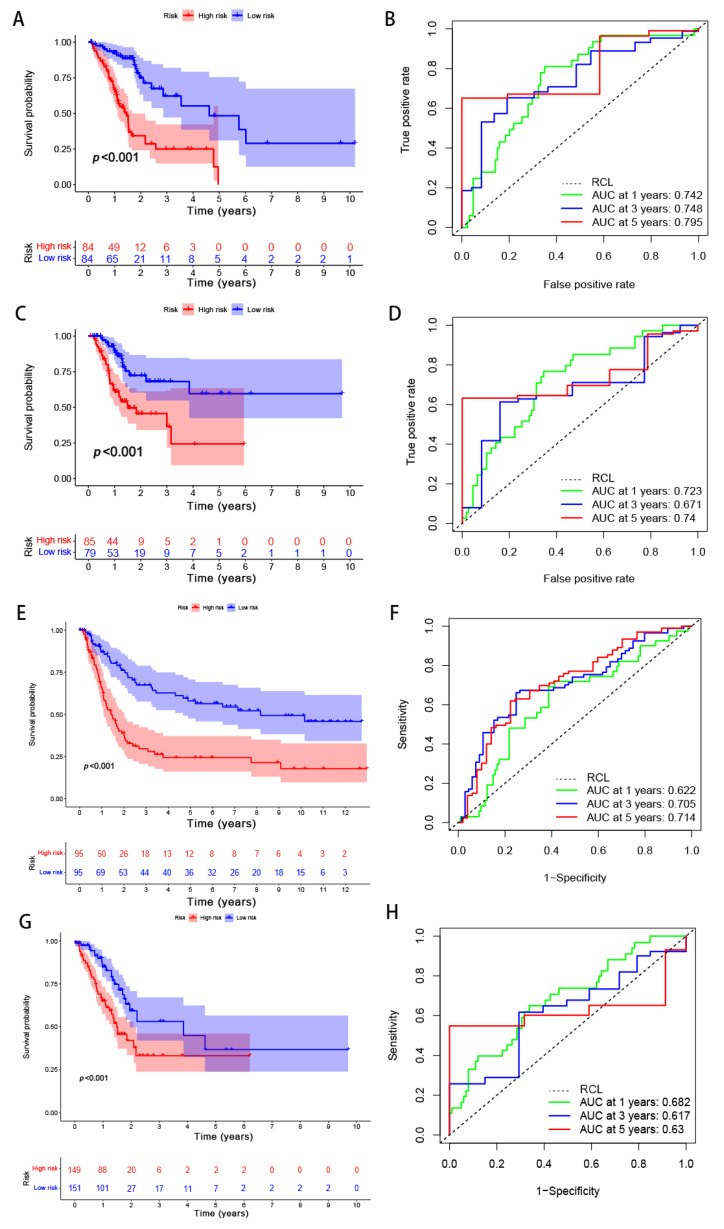
Internal and external validation of the performance regarding the predicted signatures. (**A**) The Kaplan–Meier survival curves of the training set. (**B**) The ROC curves and AUCs for the one-, three-, and five-year survival groups of the training set. (**C**) The Kaplan–Meier survival curve of the testing set. (**D**) The ROC curve and AUCs for the one-, three-, and five-year survival groups of the testing set. (**E**) The Kaplan–Meier survival curves of the GSE15459 set. (**F**) The ROC curves and AUCs for the one-, three-, and five-year survival groups of the GSE15459 set. (**G**) The Kaplan–Meier survival curves of the GSE62254 set. (**H**) The ROC curves and AUCs for the one-, three-, and five-year survival groups of the GSE62254 set. RCL: random guessing line, the baseline of a ROC curve.

**Figure 9 biomolecules-13-00469-f009:**
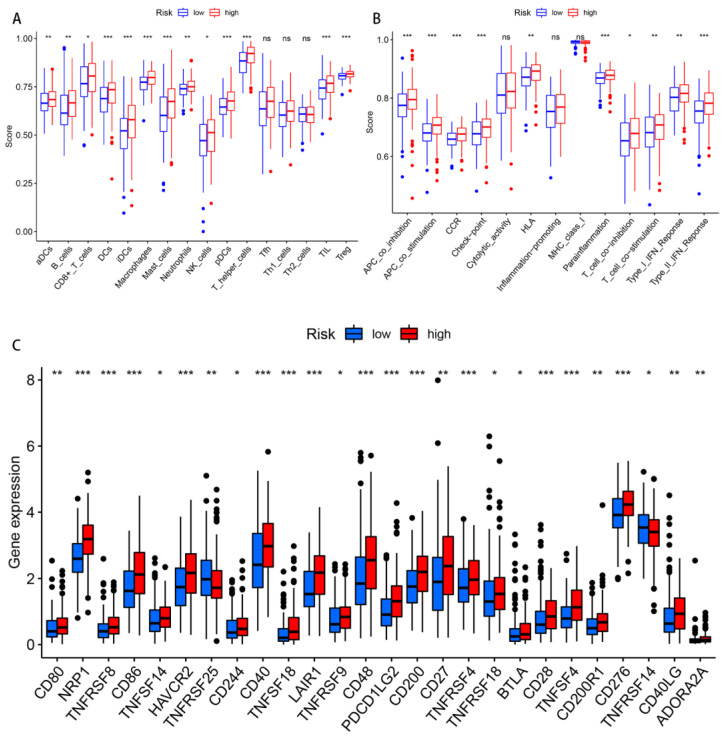
Differences in the immune-infiltrating cells and immune-related functions between the two risk groups. (**A**,**B**) The infiltration levels of the immune cells, as well as the immune-related function scores in the high- and low-risk groups. (**C**) The expression of the immune checkpoints in both risk groups. Abbreviations—ns: non-significant. * *p* < 0.05; ** *p* < 0.01; and *** *p* < 0.001.

**Figure 10 biomolecules-13-00469-f010:**
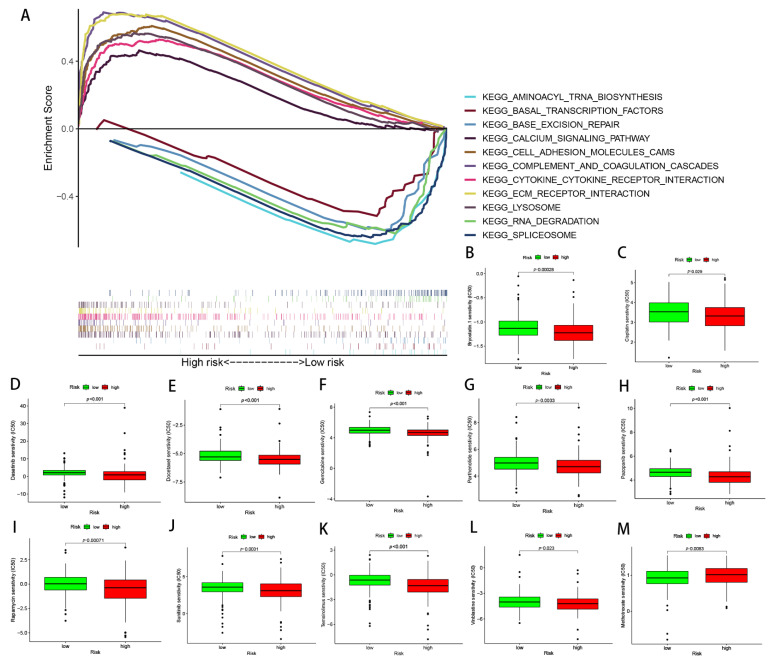
Functional enrichment and drug susceptibility analysis of the predicted signatures. (**A**) The function of pathways associated with the predicted signatures, as determined via GAEA. (**B**–**M**) The IC50 of Bryostatin.1, Cisplatin, Dasatinib, Docetaxel, Gemcitabine, Parthenolide, Pazopanib, Rapamycin, Sunitinib, Temsirolimus, Vinblastine, and Mitomycin C in the high- and low-risk groups.

**Figure 11 biomolecules-13-00469-f011:**
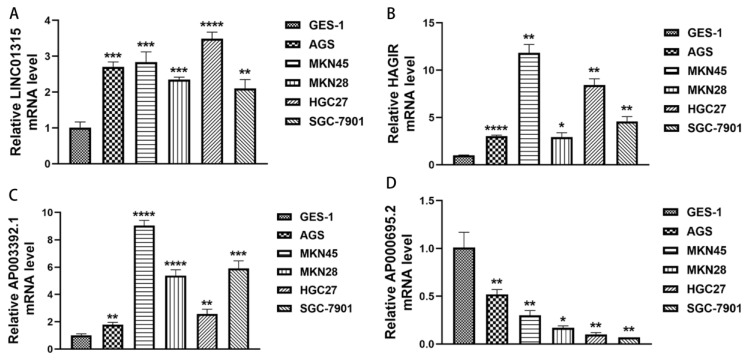
The expression levels of the PRlncRNA in the GC cell lines. (**A**) LINC01315. (**B**) HAGLR. (**C**) AP003392.1. (**D**) AP000695.2. * *p* < 0.05; ** *p* < 0.01; *** *p* < 0.001, and **** *p* < 0.0001.

**Table 1 biomolecules-13-00469-t001:** The sequences of primers used in this study.

Primers Gene	Sequence (5′–3′)Forward Primer	Reverse Primer
LINC01315	TCCGCGCTCTGAAGGATCTC	CTTGACCACCCCCGGATTC
HAGLR	TCCCCACCTTCCCCAAAGTA	GGAGGGTCTACCTCGTTTGC
AP000695.2	GGACACTCTGAAGGAACTC	GATGACCATTAGCCAACAAG
AP003392.1	GAATTCACCCACCTCAGCC	GTGTGCGTTTTCCCACTGTC
GAPDH	GGAGTCCACTGGCGTCTTCA	GTCATGAGTCCTTCCACGATACC

**Table 2 biomolecules-13-00469-t002:** The risk model of six pyroptosis-related lncRNAs with prognostic values in GC.

LncRNA	Coef	HR	HR.95L	HR.95H	*p*-Value	Risk
LINC01315	−0.44778677	0.639040932	0.397446603	1.027492273	0.064597435	Low
AL161785.1	0.581346496	1.788444945	1.078224415	2.966483855	0.024342675	High
AP003392.1	−0.673152024	0.5100982	0.303220602	0.858121684	0.011195944	Low
AP000695.2	0.564391358	1.758377234	1.065509457	2.901795454	0.027228282	High
HAGLR	0.449545667	1.567599812	1.163590693	2.111884517	0.003113121	High
AL590666.2	−0.216868363	0.805035935	0.669152194	0.968513385	0.021497057	Low

Coef: the coefficient of lncRNAs correlated with survival, HR: hazard ratio, HR.95L: low 95% CI of HR, and HR.95H: high 95% CI of HR.

**Table 3 biomolecules-13-00469-t003:** Clinical features of stomach adenocarcinoma (STAD) patients in TCGA database.

Feature	*N* (294)	%
Age (years)		
≤65	135	45.9
>65	159	54.1
Vital status		
Alive	188	63.9
Dead	106	36.1
Gender		
Female	109	37.1
Male	185	62.9
Grade		
G1	7	2.4
G2	101	34.4
G3	186	63.2
TNM stage		
Stage I	37	12.6
Stage II	97	33
Stage III	130	44.2
Stage IV	30	10.2
T stage		
T1	13	4.4
T2	60	20.4
T3	144	49
T4	77	26.2
M stage		
M0	276	93.9
M1	18	6.1
N stage		
N0	89	30.3
N1	80	27.2
N2	63	21.4
N3	62	21.1

## Data Availability

Publicly available datasets were analyzed in this study, and these can be found in the Cancer Genome Atlas (https://portal.gdc.cancer.gov/, accessed on 13 January 2022). The code that supports the findings of this study is available from the corresponding author upon reasonable request.

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
