# Peer review of "Prognosis Risk Model Based on Pyroptosis-Related lncRNAs for Gastric Cancer"

_biomolecules, 2023, doi:10.3390/biom13030469_

Round 1

Reviewer 1 Report

SEE THE ATTACHED FILE

Author Response

Responses to the reviewer's comments can be founded in the attachment.

The English-Editing-Certification and revised figures can be downloaded in the system.

Reviewer 2 Report

The manuscript focuses on the identification of a novel pathway built to define risk model for GC patients derived from lncRNA evaluation represents a reliable, technically coirrect manuscript where few minor integrations should be implemented to accept this paper for the publication

- In the introduction section, please, could the authors better define the actual considerations about molecules (including lncRNA, miRNA...) in the clinical administration of GC patients

- In the results section, the authors clearly define the statistically relevant results obtained from in vitro and in silico analysis. As regards, please, could the authors show if integrative analysis of most promising lncRNA may also represent a plus for the identification of recurrence risk for GCp patients?

- Could this model may be applied in other clinical setting? could the authors discuss this point?

Author Response

(The authors gave the same response as above.)
